

# 200-years ice core bromine reconstruction at Dome C (Antarctica): observational and modelling results

François Burgay[1,2], Rafael P. Fernandez[3], Delia Segato[2,4], Clara Turetta[2,4], Christopher S. Blaszczak-Boxe[5], Rachael. H. Rhodes[6], Claudio Scarchilli[7], Virginia Ciardini[7], Carlo Barbante[2,4], Alfonso Saiz-Lopez[8] and Andrea Spolaor[2,4*]

[1]Paul Scherrer Institute, Laboratory of Environmental Chemistry (LUC), 5232 Villigen PSI, Switzerland

[2]University Ca' Foscari of Venice, Department of Environmental Sciences, Informatics and Statistics, 30172 Venice Mestre, Italy

[3]Institute for Interdisciplinary Science, National Research Council (ICB-CONICET), FCEN-UNCuyo, Mendoza, 5501, Argentina

[4]National Research Council, Institute of Polar Sciences, 30172 Venice Mestre, Italy

[5]Department of Geosciences, The Pennsylvania State University, State College, PA 16803, United States

[6]Department of Earth Sciences, University of Cambridge, Cambridge, United Kingdom

[7]Laboratory of Observations and Measures for the Environment and Climate (SSPT-PROTER-OEM), ENEA, Rome, Italy

[8]Department of Atmospheric Chemistry and Climate, Institute of Physical Chemistry Rocasolano, CSIC, Madrid, Spain

*Corresponding author: andrea.spolaor@cnr.it

**Keywords (max 6)**: bromine, ice core, sea-ice variability, Dome C (Antarctica), ozone hole

Key points (max 3)

- Stratospheric ozone-hole depletion has not affected bromine preservation in snow at Dome C
- Volcanic eruptions have not significantly altered the snow bromine profile at Dome C over the last 200 years
- Little seasonal sea-ice variability over the last 30 years and low sensitivity to first-year sea-ice bromine emissions at Dome C do not allow the validation of $Br_{enr}$ as past sea-ice proxy at this site.

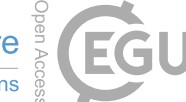

**Abstract**
Bromine enrichment (Br$_{enr}$) has been proposed as an ice core proxy for past sea-ice reconstruction.
Understanding the processes that influence bromine preservation in the ice is crucial to achieve a
reliable interpretation of ice core signals and to potentially relate them to past sea-ice variability. Here,
we present a 210-years bromine record that sheds light on the main processes controlling bromine
preservation in the snow and ice at Dome C, East Antarctic plateau. Using observations alongside a
modelling approach, we demonstrate that the bromine signal is preserved at Dome C, and it is not
affected by the strong variations in ultraviolet radiation reaching the Antarctic plateau due to the
stratospheric ozone hole. Based on this, we investigate whether the Dome C Br$_{enr}$ record can be used as
an effective tracer of past Antarctic sea-ice. Due to the limited time window covered by satellite
measurements and the low sea-ice variability observed during the last 30 years in East Antarctica, at
this stage we cannot fully validate Br$_{enr}$ as an effective proxy for past sea-ice reconstructions at Dome
C.
**1.  Introduction**

15       Halogens play an important role in the chemistry and oxidizing capacity of the Earth's atmosphere:

they take part in new particle formation processes, promote mercury oxidation, influence the budget of
HO$_x$ and NO$_x$ radicals and cause ozone depletion through efficient catalytic cycles (Saiz-Lopez and
Von Glasow, 2012; Simpson et al., 2007). Volcanic eruptions (Gutmann et al., 2018) and the ocean
(Parrella et al., 2012; Prados-Roman et al., 2015) represent the main natural sources of halogens to the
atmosphere, releasing significant amounts of bromine (Br) and iodine (I) (Cuevas et al., 2018; Carpenter
et al., 2013). In this work we focus on bromine, which has been shown to dominate halogen emissions
and chemistry in the polar atmosphere through the so-called "bromine explosion events" (Pratt et al.,
2013; Platt and Lehrer, 1997). These are heterogeneous autocatalytic photochemical reactions, which
were first described in the Arctic boundary layer and that cause the bromine-induced ozone depletion
events (ODEs) (Fan and Jacob, 1992; Vogt et al., 1996; Foster et al., 2001; Wennberg, 1999; Barrie et
al., 1988; Kreher et al., 1997). These autocatalytic multiphase chain reactions require both acidic



27 conditions and sunlight to produce an exponential increase in atmospheric bromine concentration,

28 mainly as gaseous BrO, $Br_2$ and HOBr (Schönhardt et al., 2012; Zhao et al., 2016; Nghiem et al., 2012).

29 In the polar regions, the most favourable substrate (i.e. with a large bromide content) to produce such

30 bromine explosion and ODEs during springtime is the sea-salt aerosol derived from surface blowing

31 snow deposited over first-year sea-ice (FYSI), which is characterized by acidic conditions, higher $Br^-$

32 /$Cl^-$ ratio (Pratt et al., 2013) and higher salinity than the snow deposited over multi-year sea-ice (MYSI)

33 (Frey et al., 2020). Direct observations from two winter cruises in the Weddell Sea (Antarctica) and

34 model simulation, showed that significant bromine losses take place in the aerosol phase, indicating

35 that sea-salt aerosol debromination from salty blowing snow over sea-ice represents a relevant source

36 of gas-phase inorganic bromine to the troposphere (Frey et al., 2020; Parrella et al., 2012; Yang et al.,

37 2005). Note that most inorganic bromine gases present in the atmosphere are highly water-soluble and

38 they suffer wet and dry deposition over the ice sheets (Legrand et al., 2021; Parrella et al., 2012;

39 Fernandez et al., 2019).

40  Especially during the "bromine explosion events", bromine, that shares the same sources as sodium

41 (i.e. sea-salt), is significantly enriched compared to sodium (Na) in the FYSI surfaces, exceeding the

42 bromine-to-sodium mass ratio of seawater  (Millero et al., 2008). As discussed above, this enrichment

43 is mainly promoted by the presence of FYSI compared to MYSI, thus, bromine enrichment ($Br_{enr}$, eq 1)

44 has been proposed as a potential tracer for the reconstruction of past FYSI conditions (Spolaor et al.,

45 2016; Maffezzoli et al., 2019; Spolaor et al., 2013b; Vallelonga et al., 2017), with higher $Br_{enr}$ values

46 corresponding to larger FYSI extent (Spolaor et al., 2013b). However, many unknowns, mainly related

47 to the source, transport and preservation of bromine within the snowpack, still remain (Maffezzoli et

48 al., 2019). For example, it has been suggested that the anthropogenic-induced acidity increase of the

49 snow deposited over the sea-ice surface can enhance the sea salt debromination rates, thus, enhancing

50 the release of reactive bromine from sea-salt aerosols into the atmosphere (Maselli et al., 2017; Sander

51 et al., 2003). Further, it is possible that bromine can also be re-emitted from the snowpack after

52 deposition and prior to burial. However, the results obtained from previous studies are contradictory

53 and site-specific (Mcconnell et al., 2017; Legrand et al., 2016; Dibb et al., 2010; Spolaor et al., 2019).



In Greenland, Dibb et al., (2010) showed that bromine photo-activation was present during
spring/summer and highlighted an efficient Br chemical cycling above the snow. In the Svalbard
Archipelago, a high-temporal resolution study designed to investigate the potential photo-emission of
bromine from the snowpack (Spolaor et al., 2019) did not highlight any bromine diurnal cycle,
suggesting its preservation in snow. In Antarctica, through the investigation of Na and Br fluxes against
snow accumulation rate, McConnell et al., (2017) found that bromine re-emission from the Antarctic
snowpack is linearly dependent on the accumulation rate. It was shown that the bromine loss from the
snowpack was higher (65%) at sites with the lowest accumulation rate (50 kg m$^{-2}$ yr$^{-1}$), and it decreased
to 11% at sites with high annual accumulation rate (300 kg m$^{-2}$ yr$^{-1}$). Based on these observations,
virtually all bromine deposited at Dome C (Antarctica), where the annual snow accumulation is ≈ 25
kg m$^{-2}$ yr$^{-1}$, would have been re-emitted to the atmosphere prior to burial (Mcconnell et al., 2017;
Maffezzoli et al., 2019). These conclusions contrast with previous observations performed at Dome C
that reported no significant bromine re-emission from the snowpack (Legrand et al., 2016). An
additional process that can affect bromine preservation within the snowpack has been identified in
coincidence with the 17.7 ka Mt. Takahe volcanic eruption, when, the combination of an increased
surface ultraviolet (UV) radiation, due to stratospheric ozone depletion, and high acidity conditions
were associated with a decrease in ice bromine concentration (Mcconnell et al., 2017). To our
knowledge, there are no investigations that focused on the effects of the modern changes on UV-
radiation reaching the Antarctic plateau surface due to the ozone-hole formation on bromine
preservation in snow.

To unravel the physicochemical processes that can influence bromine preservation in the snow, we

investigated the main pathways that can induce its emission from the snowpack to the atmosphere.
Bromine is mainly present in the Antarctic snowpack as bromide (Spolaor et al., 2013a) and it can be
oxidized by OH radicals (George and Anastasio, 2007; Abbatt et al., 2010) to form evaporable gaseous
bromine. The main •OH sources within the snowpack are the photolysis of hydrogen peroxide (Chu and
Anastasio, 2005), nitrate (Chu and Anastasio, 2003; Abbatt et al., 2010) and nitrite (Chu and Anastasio,
2007). Understanding the relevance of each of these photochemical pathways in explaining the



preservation of bromide in the Antarctic snowpack before and after the onset of the modern-ozone hole
is then crucial for a reliable interpretation of the bromine enrichment profile observed in ice core
records.

In this study, we present the first bromine record retrieved from a shallow firn core collected at

Dome C, Antarctica, covering the period 1800-2012. Through the evaluation of the bromine profile, it
is possible to provide new evidence about the role of the enhanced solar-UV radiation due to the onset
of the modern Antarctic ozone hole (1975) on bromine preservation in the snowpack. An extended
evaluation of the role of $^{\bullet}$OH precursors and their relevance at Dome C is also performed. Lastly, an
assessment of the possibility of using $Br_{enr}$ as a potential past sea-ice tracer at Dome C is also addressed
by combining re-analysis of air-mass transport and satellite observations of sea-ice extent. The results
presented in this paper open new perspectives on future long-term bromine studies from low
accumulation rate ice cores, aimed at forecasting future deep core drillings at Dome C, such as those
planned for the Beyond EPICA project.
**2.  Material and Methods**

*2.1 Ice core sampling and location*


A 13.72 m shallow ice-core was drilled close to Concordia Station, at Dome C (3233 m a.s.l.;

75°05'59''S, 123°19'56''E) in 2012. This record covers approximately 212 years, from 1800 to 2012.
The ice core dating is described in detail by Spolaor et al., (2021). Dome C is a suitable Antarctic site
for performing photochemical studies related to the preservation of reactive elements and halides within
the snowpack (Savarino et al., 2007; Cairns et al., 2021; Song et al., 2018; Spolaor et al., 2018; Spolaor
et al., 2021). This location presents a low and rather constant accumulation rate ($25 \pm 1.3$ kg m$^{-2}$ yr$^{-1}$
from 1816-1998, $26 \pm 1.3$ kg m$^{-2}$ yr$^{-1}$ from 1955-1998; 27 kg m$^{-2}$ yr$^{-1}$ from 2006-2013) (Frezzotti et al.,
2005), and it is located about 1000 km away from shorelines, thus not being directly affected by local
coastal emissions. Matching these criteria is essential for the evaluation of the effects of modern
stratospheric ozone loss due to long-lived ozone depleting substances in the potential bromine release
from the snowpack.



The shallow ice core was collected using a hand drill (3 inches diameter); the sections were sealed
in plastic containers and shipped to the Institute of Polar Sciences of the National Research Council
(ISP-CNR) in Venice. The ice core sections were subsequently sampled at 5 (± 1) cm resolution using
a ceramic knife, rinsed with Ultra-Pure Water (UPW, Elga Lab, UK) after each use. Only the central
part of the core was collected into 50 mL pre-cleaned polyethylene (PE) vials for subsequent analyses,
while the outer 2 cm were removed by scraping with a ceramic knife. The core samples were processed
in a class 1000 inorganic clean room under a class 100 laminar-flow bench. Samples were kept at -20°C
and under dark conditions until the analysis to avoid any possible photolysis reaction. Bromine and
sodium analyses were conducted on melted and not acidified samples by Inductively Coupled Plasma—
Sector Field Mass Spectrometry (ICP-SFMS) (see 2.2). The sodium record considered in this study had
some gaps from 1989 to 1997 (n=12, corresponding to 5% of the total amount of samples) that were
filled with Na concentration data retrieved from two snow-pits collected in 2013 and 2017, as reported
in Spolaor et al., 2021.

*2.2 Instrumental analysis and cleaning procedure*

Total sodium and bromine concentrations were determined by ICP-SFMS following Spolaor et al.,
(2016). Each analytical run started and ended with an Ultra-Pure Water (UPW) cleaning session of 3
min to ensure a stable background level throughout the analysis. The external standards that were used
to calibrate the analytes were prepared by diluting a 1000 ppm stock IC (ion chromatography) standard
solution (TraceCERT® purity grade, Sigma-Aldrich, MO, USA). The standard concentrations ranged
between 1 and 200 ng g$^{-1}$ for sodium and 0.05 and 0.200 ng g$^{-1}$ for bromine. Precision and accuracy of
the measurements were determined through the multiple reading of selected ice samples and external
standards, respectively. The relative standard deviation (RSD %) was low for all the analytes, ranging
between 3–4% for sodium and 5-7% for bromine, while accuracy, expressed as the ratio between the
observed and the true value, was 105% for sodium and 92% for bromine. The instrumental limit of
detection (LoD), calculated as three times the standard deviation of the blank (n = 10), was 1 ng g$^{-1}$ for
Na and 0.05 ng g$^{-1}$ for Br.



All the plastic material used for sample storage and analysis was washed 5 times using UPW and
filled with UPW for 1 week. Then, it was rinsed again 5 times with UPW and dried under class-100
laminar flow hood before use.

*2.3 Back-trajectories calculation and satellite observations*

Backward air mass trajectories that reach the Dome C site (75°05'59''S, 123°19'56''E) were
calculated to identify the most likely ocean and sea-ice areas that release bromine species to the
atmosphere and that can be transported to the interior of Antarctica. Back-trajectories were obtained
from the Hybrid Single-Particle Lagrangian Integrated Trajectory (HYSPLIT) model (Stein et al., 2015)
using European Centre for Medium-range Weather Forecasts (ECMWF) ERA5 meteorological
reanalysis (Hersbach et al., 2020). ERA5 is available on 37 pressure levels with a regular spatial grid
of 0.25° x 0.25° at hourly temporal sampling and is publicly available for download at the Copernicus
Climate Data Store (https://cds.climate.copernicus.eu/cdsapp#!/home). However, due to the huge
amount of trajectories needed for this study, for computation requirements, we considered ERA5
parameters on a spatial grid of 0.5° x 0.5° every three hours and 24 pressure levels (Becagli et al., 2022).
5 days backward trajectories were calculated every 3 hours at 1000, 2000 and 3000 meters above Dome
C model terrain height for the period 1979 – 2018. Each back trajectory was then projected on the Sea-
Ice Concentration field (SIC) and the 10-m wind field (still in the ECMWF ERA5 reanalysis),
associating each value along the trajectory path with the nearest SIC and wind speed values. The main
paths of air masses reaching Dome C were highlighted dividing the southern hemisphere in a regular
1° x 1° mesh and counting the total number of back trajectories points at 5 days, falling in each grid cell
(i.e. the hours spent by the air mass in each grid cell). Since bromine species are emitted in the marine
boundary layer (MBL) (Sander et al., 2003), source bromine areas were evaluated, selecting only the
trajectory where the air mass paths lie within the MBL and over sea-ice and counting the total number
of resulting points, where conditions are fulfilled, in each of the 1° x 1° grid cells. The height of the
MBL was set equal to the 900 hPa Isosurface (Lewis et al., 2004) and a value of SIC > 15%  were
considered in order to simulate the presence of the sea-ice cover (Becagli et al., 2022). More details



about the computation of different back-trajectories as a function of different heights are provided in
the Supplementary Material (Figure S1).
The sea-ice concentrations used in this work are derived from passive-microwave radiometers on
NASA's satellites. The data are publicly available at the NASA Earth Science portal
(https://earth.gsfc.nasa.gov/) and at the National Snow and Ice Data Center portal (http://nsidc.org). The
sea-ice extents (in $km^2$) are calculated as the hemispheric total as well as five regions in the Southern
Ocean (Indian Ocean, Western Pacific, Ross Sea, Bellingshausen and Amundsen seas and the Weddell
sea).
*2.4 CAM-CHEM model set-up*
The wavelength-dependent solar ultraviolet (UV) radiation reaching the Antarctic plateau
surface at Dome-C during the 1950-2010 period was computed using the Community Earth System
Model (CESM) (Tilmes et al., 2016). The setup of the atmospheric component of the model (CAM-
Chem, version 4) was identical to the one used in previous studies addressing the evolution of iodine
icecore records in the Arctic (Cuevas et al., 2018) and Antarctica (Spolaor et al., 2021), and considers
prescribed sea surface temperatures and sea-ice distributions following the CCMI-REFC1
recommendation (Eyring et al., 2013). The CAM-Chem very short-lived (VSL) setup includes
geographically distributed and seasonally–dependent natural oceanic emissions of five bromocarbons
($VSL^{Br}$ = $CHBr_3$, $CH_2Br_2$, $CH_2BrCl$, $CHBrCl_2$, $CHBr_2Cl$) and four iodocarbons ($VSL^I$ = $CH_3I$, $CH_2ICl$,
$CH_2IBr$, $CH_2I_2$), whose oceanic flux is assumed to remain constant during the whole modelling period
(Ordóñez et al., 2012). The chemical scheme includes the additional inorganic chlorine and bromine
contribution arising from the so-called sea-salt dehalogenation recycling occurring in the marine
boundary layer and the free troposphere (Fernandez et al., 2014; Fernandez et al., 2021). The model
was configured in free-running mode, with 26 vertical levels expanding from the Earth's surface to
approximately 40 km (3.5 hPa in the upper stratosphere), and with a spatial resolution of 1.9º latitude
by 2.5º longitude. The CAM-Chem CCMI-REFC1 configuration used here provides a reasonable
representation of the evolution of the size and depth of the ozone hole, presenting an excellent



agreement with satellite ozone observations during the modelled period (Fernandez et al., 2017; Spolaor
et al., 2021).

Here, we compute the photolysis rate constant (J-value) of different ˙OH precursors involved in the

preservation of bromine in the snowpack following the same approach used in Spolaor et al., (2021).
The molar absorptivities, quantum yields and species concentrations for hydrogen peroxide, nitrate and
nitrite at Dome-C are summarized in Table 1. The modeled actinic flux reaching the Antarctic surface
includes 100 bins expanding from 121 nm to 750 nm, with a spectral resolution ranging from less than
1 nm in the UV to 50 nm in the visible edge. In particular, the model bandwidth within the 280-400 nm
spectral range considered in this work poseess a mean resolution of 4 nm. Thus, the CAM-Chem surface
actinic flux for each wavelenthg grid was linearly interpolated into a 1 nm bandwidth, and the mean J-
value during the whole sunlit period (i.e., from September 1$^{st}$ of a given year to February 28$^{th}$ of next
year) was computed offline at the closest gridbox to Dome C (74.84° S; 122.5°E; model mean altitude
of 3300 m a.s.l.). The complete sunlit period (spring + summer) was selected because even when the
largest changes in surface actinic flux associated with the ozone hole formation are observed during
spring, the UV radiation intensity reaching the Antarctic plateau maximizes during the summer (Spolaor
et al., 2021).
**3    Results and discussion**

*3.1 Sodium, bromine and bromine enrichment profiles from the Dome C shallow core*

Dome C is in the Antarctic plateau, at approximately 1000 km from the coast and 3233 m a.s.l. Air

masses arriving at the site originate from a vast area that extends over the Eastern Antarctic Ocean,
Ross Sea and in minimal percentage from West Antarctica. Back trajectory analysis (Figure S1)
confirms that the most likely source areas of bromine and sodium emissions during the 1979-2018
period extend from the Indian Ocean sector (IO, 11%) up to the Ross Sea sector (RS, 21%), with the
most likely area being the Western Pacific sector (WP, 45%), with the remaining 17% from
Bellingshausen and Amundsen seas sector.





The variability in sodium concentration, a conservative tracer that does not show any post-
depositional transformation, is used here to evaluate the marine contribution at Dome C (Caiazzo et al.,
2021). Sodium concentrations along the entire record spanned from 12 to 117 ng g$^{-1}$, with an average
value of 40 ± 13 ng g$^{-1}$. Its profile (Figure 1) shows an average value of 39 ± 13 ng g$^{-1}$ from 1800 to
1994, while it showed a significant increase over the last 18 years of the record (50 ± 9 ng g$^{-1}$) suggesting
an enhanced transport towards Dome C. Bromine concentration at Dome C ranges from below the LoD
(0.05 ng g$^{-1}$) to 0.41 ng g$^{-1}$, with an average value of 0.10 ± 0.05 ng g$^{-1}$ along the entire record. A
significant bromine increase was detected since 2004 when the concentration increased from 0.10 ±
0.05 ng g$^{-1}$ (pre-2004) to 0.23 ± 0.09 ng g$^{-1}$ (post-2004). The abrupt changes both in the Na and Br
signals were detected using the *findchangepts* Matlab function based on an optimal detection of
changepoints method. The absence of an abrupt change of the bromine signal at the onset of the ozone
hole (1975), indicates that it can be preserved in the snowpack independently on the incoming UV-
radiation (see section 3.2). Sodium and bromine did not show any significant correlation (r = 0.06, *p*-
value = 0.20) along the entire record, suggesting different deposition velocities during transport from
the coast, with sodium being deposited faster than bromine. Indeed, in the polar atmosphere, sodium is
present in the aerosol phase, it is mainly subjected to wet deposition processes and its concentration
decreases significantly with distance from the coast, reaching a rather constant deposition rate at
approximately 400 km inland. Contrarily, bromine exists in both the aerosol and in the gas phase, its
atmospheric lifetime is also driven by dry deposition processes, it can experience heterogeneous
chemical recycling during transport and its concentration quasi-linearly decreases from 100 km to 1000
km inland (Simpson et al., 2005; Vallelonga et al., 2021).
The bromine enrichment values in the ice samples were calculated as:

$$Br_{enr} = \frac{[Br]}{[Na]} \cdot 0.0062 \qquad \text{(eq. 1)}$$

where Br and Na are the concentrations obtained from the Dome C record and 0.0062 reflects the
bromine-to-sodium mass ratio in seawater (Spolaor et al., 2013b; Millero et al., 2008). Br$_{enr}$ values
ranged between 0.07 and 1.6 and, contrarily to sodium and bromine, the enrichment did not show any





significant increase in the recent part of the record. On the contrary, a significant regime change was
detected in 1825, when the $Br_{enr}$ mean value changed from $0.7 \pm 0.3$ (1800-1825) to $0.4 \pm 0.3$ (1825-
2012). A two-sample *t*-test strengthened the significant difference between the two periods (*p*-value <
.001). In general, $Br_{enr}$ values were mainly below 1 (i.e. bromine is depleted relative to sodium), which
is expected at remote locations like Dome C, since the Br to Na ratio depends on the relative transport
times of sea salt aerosol and gaseous bromine compounds in the atmosphere (Spolaor et al., 2013b;
Simpson et al., 2007; Vallelonga et al., 2021). Thus, our findings agree with the synthesis of Vallelonga
et al. (2021), which shows $Br_{enr}$ decreases as function of the distance to the coastal source, where $Br_{enr}$
can reach the value of 60, and that is depleted in locations far from the source. The few $Br_{enr}$ values
higher than 1 may indicate larger FYSI surface from the source areas (Spolaor et al., 2016) or different
bromine partitioning between the aerosol and gas phase that depends on the aerosol size and,
consequently, on atmospheric resident times (Legrand et al., 2016; Maffezzoli et al., 2019; Vallelonga
et al., 2021). In addition, changes in background atmospheric $^{\cdot}OH$ and $NO_x$ might have had an impact
on the gas-phase bromine partitioning between reservoir and reactive species that might have led to a
faster bromine deposition since species like $BrONO_2$ and HOBr have larger and more efficient
deposition velocities than reactive species like BrO and Br (Saiz-Lopez and Fernandez, 2016;
Fernandez et al., 2019).
In Antarctica, few other long-term bromine records exist and they were all collected from coastal
sites (Spolaor et al., 2013b; Vallelonga et al., 2017). In contrast to Dome C, these records are more
directly influenced by local marine contributions and a shorter atmospheric transport time from the
source to the deposition location, which is reflected by the higher $Br_{enr}$ values (Vallelonga et al., 2017).

*3.2 Bromine preservation in the snowpack at Dome C*

When reaching the snowpack, UV radiation can rapidly break weak chemical bonds and, due to its
high energy, can promote photochemical reactions, especially in the UV-A (320–400 nm) and UV-B
(290-320 nm) regions (Grannas et al., 2007). Due to its low accumulation rate, Dome C is the perfect
location for performing UV-photolysis studies on chemical species occurring at the ice and snow



surface (Frey et al., 2009; Savarino et al., 2007). The stratospheric ozone layer depletion observed since
1975, has caused an increase in the incoming solar UV radiation over Antarctica at $\lambda < 300$ nm,
enhancing, for example, the photochemical iodide oxidation and its subsequent release from the
snowpack (Spolaor et al., 2021).

As highlighted in section 3.1, no significant changes neither in bromine concentration nor in

bromine enrichment have been detected at Dome C since 1975, suggesting that, contrarily to iodine, the
enhanced UV-radiation reaching the Antarctic plateau has not altered bromine preservation within the
snowpack (Figure 1). Moreover, lab and chamber experiments showed enhanced photochemical
oxidation and subsequent release of $I_{2(g)}$ from artificial snow/ice and the snowpack through the
formation of a critical I-$O_2$ complex having an absorption band centred at 290 nm (Kim et al., 2016).
At present, there is no evidence and/or available literature describing a similar Br-$O_2$ complex, and/or
any other brominated intermediate product, that leads then to the release of $Br_{2(g)}$. In fact, the main
inorganic route for bromide oxidation requires radical oxidants (e.g. $^\bullet OH$) to drive the redox production
of hypobromous acid (BrOH) (Artiglia et al., 2017). This oxidized species can then combine with other
reduced halide ions to form molecular halogen compounds that are released into the gas phase (eq. 2-
5) (George and Anastasio, 2007):

$$Br^- + {}^\bullet OH \rightarrow {}^\bullet BrOH^- \tag{2}$$

$$^\bullet BrOH^- + Br^- \rightarrow {}^\bullet Br_2^- + OH^- \tag{3}$$

$$^\bullet Br_2^- + HO_2{}^\bullet \rightarrow Br_2 + HO_2^- \tag{4}$$

$$^\bullet Br_2^- + {}^\bullet Br_2^- \rightarrow Br_2 + 2Br^- \tag{5}$$

Over ice and snow substrates, hydroxyl radicals ($^\bullet OH$) can be produced by the photolysis of

hydrogen peroxide ($H_2O_2$), nitrate ($NO_3^-$) and nitrite ($NO_2^-$) (Chu and Anastasio, 2005; Abbatt et al.,
2010; Chu and Anastasio, 2007). Between 290-340 nm, $H_2O_2$ has a wavelength-dependent molar
absorptivity that is 2.5-7.1 times lower than that for $NO_3^-$. Nevertheless, $H_2O_2$ has a $\approx$ 160 times greater
quantum yield for $^\bullet OH$ production (Table 1) that is insensitive to ionic strength, pH and wavelength
(Chu and Anastasio, 2005). Therefore, for a given concentration, $H_2O_2$ is a much more effective source





of $^\bullet$OH than nitrate. To our knowledge, the only $H_2O_2$ concentration value available at Dome C is 2 ng
$g^{-1}$, derived from a sample collected at 3.5 m-depth (Frey et al., 2006). This low value, compared to
other locations, is consistent with semi-empirical models that predict a complete hydrogen peroxide
loss when the accumulation rate is below $\approx$ 70 kg m$^{-2}$ yr$^{-1}$ and the annual mean temperature is -50°C
(Frey et al., 2006). Considering that Dome C has an annual mean accumulation of $\approx$ 25 kg m$^{-2}$ yr$^{-1}$
(Genthon et al., 2016) and an annual mean temperature between -54 and -50°C (Genthon et al., 2021),
we assume that the majority of the deposited or *in situ* produced $H_2O_2$ is rapidly lost to the atmosphere.
Alternatively, $NO_3^-$ photolysis (Table 1), occurring at wavelengths of 290-340 nm, with a maximum
at 320 nm (Winton et al., 2020), can act as a $^\bullet$OH source following the equations 6-9 (Chu and Anastasio,
2005; Abbatt et al., 2010; Boxe, 2005):

$$NO_3^- + h\nu \rightarrow NO_2^- + O^- \tag{6}$$

$$O^- + H_2O \rightarrow {}^\bullet OH + OH^- \tag{7}$$

$$H^+ + NO_2^- \rightarrow HONO \tag{8}$$

$$HONO + h\nu \rightarrow {}^\bullet OH + NO \tag{9}$$

The $^\bullet$OH radicals, formed by nitrate photolysis can produce $Br_{2(g)}$ following reactions 2-5. The
typical snowpack nitrate profile at Dome C ranges between 22 and 147 ng g$^{-1}$ (Caiazzo et al., 2021;
Spolaor et al., 2021) and shows an exponential decay in concentration driven by nitrate UV-photolysis
and recycling (Winton et al., 2020; Röthlisberger et al., 2000; Savarino et al., 2007). Due to its higher
concentration compared with hydrogen peroxide, nitrate may represent a relevant $^\bullet$OH source at Dome
C despite its lower quantum yield for $^\bullet$OH production. The nitrate UV-photolysis, followed by $^\bullet$OH
formation and $Br_{2(g)}$ emission, has been reported under laboratory conditions with a significant
dependency on the ice pH, with the largest $Br_{2(g)}$ emissions observed at low pH (George and Anastasio,

2007).

Another source of OH radicals is nitrite (Minero et al., 2007), which can be produced by the
dissociation of nitrous acid (HONO) in the condensed phase (eq. 10);





$$HONO \leftrightarrow NO_2^- + H^+ \tag{10}$$

or by the direct formation from the minor channel (quantum yield of approximately 0.0011) of nitrate
photolysis (eq. 6) (Dubowski et al., 2002).

Nitrite displays two major absorption bands peaking at 300 and 354 nm and, through its

photolysis, it can produce $^{\cdot}$OH (eq. 11-12):

$$NO_2^- + h\nu \rightarrow NO + {^{\cdot}O^-} \tag{11}$$

$${^{\cdot}O^-} + H_2O \rightarrow {^{\cdot}OH} + OH^- \tag{12}$$

The $^{\cdot}$OH quantum yield from nitrite photo-dissociation depends both on the wavelength (increases
with decreasing wavelength) and on temperature (decreases with decreasing temperature). In addition,
nitrite has a $\approx$ 2-fold higher molar absorptivity than nitrate between 280 and 300 nm (Chu and Anastasio,
2007) and its $^{\cdot}$OH quantum yield in ice is equal to 0.020 (at 240 K, $\lambda$ = 300), which is 6-fold higher than
the one calculated for nitrate (Table 1). Unfortunately, there are no direct measurements of nitrite at
Dome C, meaning that its concentration needs to be estimated. Following the approach used by Chu
and Anastasio (2007), we assumed that at Dome C the nitrite concentration is like the one calculated at
the South Pole, which is 0.092 ng g$^{-1}$. This assumption is based on the similar $NO_3^-$ concentration
recorded both at the South Pole (99 ng g$^{-1}$) and at Dome C (90-147 ng g$^{-1}$), on the use of $NO_3^-$ photolysis
as the main source for nitrite in the snow (Chu and Anastasio, 2007) as well as the total UV-radiation
reaching both locations present similar intensities and seasonality.
To evaluate the relevance of these processes on the OH radical production in the snow-grains and,
consequently, their role in promoting $Br_{2(g)}$ emission from the snowpack, we modelled the hydrogen
peroxide, nitrate, and nitrite photo-activation before (1950-1975) and after the ozone hole formation
(post-1975) at Dome C, following the wavelength-dependent CAM-Chem model actinic flux output
and the methodology described in Spolaor et al.,2021. Both $H_2O_2$ and $NO_3^-$ exhibited a small, but
significant, enhancement on their surface photolysis (J-value) after the onset of the ozone hole ($\approx$20%
increase) due to the higher actinic flux reaching the surface at $\lambda$ < 300 nm, where most of the $H_2O_2$
absorption occurs, and to a limited extent also $NO_3^-$ (Figure 2, 3). In contrast, $NO_2^-$ does not show a



significant trend on their J-values because both absorption bands maximize at longer wavelengths,
within a spectral region that is not directly affected by the formation of the ozone hole (Figure 2,3). It
is important to notice that the normalized photolysis ratio between the ozone hole period and the pre-
ozone hole period, strongly depends on the wavelength range considered to compute the J-value
integration. For example, the ratio between the ozone hole and the pre-ozone hole periods for $H_2O_2$
ranges from a minimum value of 1.2 (280-390 nm) to a maximum value larger than 5 (280-300 nm)
(Figure S2). Equivalent results are obtained for the J-value enhancements of $NO_3^-$ and $NO_2^-$ (Figure
S2), although for these species, with the strongest absorption at longer wavelengths, the upper
bandwidth limit used to perform the integration should not be located at values below 310-320 nm,
which result in an up to 3-fold increase on the normalized ratio during the pre-ozone hole period. For
these reasons, and based on the observed molar absorptivities of each species shown in Figure 2a, our
best estimate of the normalized photolysis ratio shown in Figure 3c was computed considering the
following wavelengths ranges (see vertical dashed lines in Figure 2a) 280-378 nm for $H_2O_2$, 295-357.5
nm for $NO_3^-$ and 280-390 nm for $NO_2^-$.

Taking into consideration the $NO_3^-$, $H_2O_2$, and $NO_2^-$ concentrations at Dome C and the $^\bullet OH$ quantum

yields from their photolysis (Table 1), we observed that the rates of $^\bullet OH$ formation especially from the
photolysis of $H_2O_2$ and $NO_3^-$ slightly increased by a factor 1.1 and 1.09, respectively, compared to the
pre-ozone hole period, while for nitrite the enhancement was almost negligible (1.01). The average
contribution of each of this species in producing OH radicals in the snow was the same both before and
after the formation of the ozone hole, that is 69% from $NO_3^-$, 23% from $H_2O_2$, and 8% from $NO_2^-$ (Figure
3). Specifically, the $^\bullet OH$ formation rates during the ozone-hole (pre-ozone hole) period from $NO_3^-$, $H_2O_2$
and $NO_2^-$ photolysis are equal to $1.14E^{-13}$ ($1.05E^{-13}$) M s$^{-1}$, $3.80\ E^{-13}$ ($3.47E^{-13}$) M s$^{-1}$ and $1.2E^{-14}$ ($1.2E^{-14}$)
M s$^{-1}$, respectively (Figure 3). Those values are calculated by multiplying the mean photolysis rate
constant for $^\bullet OH$ formation during the  ozone-hole (pre-ozone hole) period (i.e. that is $6.4E^{-8}$ ($5.9E^{-8}$) s$^{-}$
$^1$ for $NO_3^-$,$5.9\ E^{-6}$ ($5.36E^{-7}$) s$^{-1}$ for $H_2O_2$ and $6.2\ E^{-5}$ ($6.2E^{-6}$) s$^{-1}$ for $NO_2^-$) by the estimated or real snow-
grain concentration (110 ng g$^{-1}$ or $1.77E^{-6}$ M, for $NO_3^-$, 2.2 ng g$^{-1}$, or $6.47E^{-8}$ M, for $H_2O_2$ and 0.092 ng
g$^{-1}$, or $2E^{-9}$ M, for $NO_2^-$). Our results are different from those computed at Neumayer station (Chu and



Anastasio, 2007), where the dominant contributor to $^{\bullet}OH$ production was $H_2O_2$ ($2.3E^{-11}$ M s$^{-1}$), followed
by $NO_3^-$ ($3.9E10^{-13}$ M s$^{-1}$) and $NO_2^-$ ($1.8$ $E10^{-13}$ M s$^{-1}$). Further, the $^{\bullet}OH$ production rate at Dome C for
$H_2O_2$ is 2-orders of magnitude lower than at Neumayer station, while it is similar for $NO_3^-$. The
contribution of nitrite at Dome C is one order of magnitude lower than that computed at Neumayer
station, where nitrite has been already considered as an insignificant source of $^{\bullet}OH$ because of its very
low estimated concentration (Chu and Anastasio, 2007). Overall, we can conclude that the contribution
of $H_2O_2$, $NO_3^-$ and $NO_2^-$ in forming OH radicals is low at Dome C both before and after the ozone hole
period and their change in photolysis is unlikely to have affected bromine preservation within the
snowpack. This is in agreement with previous empirical observations (Legrand et al., 2016). We then
propose that bromine release into the atmosphere can be favoured only in those locations where high
snow acidity (e.g. in correspondence to a volcanic horizon) and high concentration of $^{\bullet}OH$ precursors
(e.g. $H_2O_2$) are found.

*3.3 The role of volcanic eruptions in bromine preservation at Dome C*

Volcanic eruptions are a significant halogen source with the emission of large amounts of HCl, HF

and HBr (Pyle and Mather, 2009). In particular, BrO formation through heterogeneous photochemical
reactions was detected in a volcanic plume where local $O_3$ destruction occurred (Von Glasow et al.,
2009). However, the role of volcanic eruptions in affecting bromine concentration in ice and snow has
been poorly addressed. Studies performed in the European Alps (Legrand et al., 2021) and in the West
Antarctic Ice Sheet Divide (Mcconnell et al., 2017), showed opposite results, with recorded bromine
increase and depletion in coincidence with volcanic events, respectively. The Dome C shallow core
presented in this work covers at least seven past volcanic eruptions that were identified in other snow-
pits and deep cores using both $nssSO_4^{2-}$ and Fe(II) as volcanic proxies (Castellano et al., 2005; Burgay
et al., 2021; Gautier et al., 2016): Pinatubo/Cerro Hudson (1991, VEI = 6), Agung (1963, VEI = 5),
Krakatua (1886, VEI = 6), Cosiguina (1835, VEI = 5), Tambora (1815 = 7) and Unknown (1809, VEI
$\geq$ 5). VEI stands for Volcanic Explosivity Index, a commonly used quantity to define the magnitude of
a volcanic eruption (Newhall and Self, 1982). Its values range from 0 (Hawaiian eruption) to 8 (Ultra-
Plinian eruption). In this record, we did not detect any clear fingerprint neither as bromine increase nor



as depletion compared to the adjacent periods, suggesting a negligible role of volcanic eruptions in
affecting the Br snow chemistry in the inland Antarctic plateau (Figure S3). Even though our results
indicate that volcanic deposition does not affect the bromine signal at this location, in other Antarctic
locations (e.g. West Antarctic Ice Sheet Divide and Byrd cores) a bromine depletion was observed in
coincidence with the Mt. Takahe volcanic eruption (Mcconnell et al., 2017). We suggest that the reasons
behind these discrepancies can stem from different glaciochemical properties of the snow between the
two locations that need to be deeply investigated in future studies. Alternatively, these differences might
also be explained by differences in atmospheric transport from the source to the deposition locations.
3.4 *Can Br$_{enr}$ at Dome C be used as proxy for past sea-ice extent?*
Having presented evidence to demonstrate the preservation of bromine in the snowpack at Dome C
and the absence of a link between our Br$_{enr}$ signal and the formation of the ozone hole or volcanic
eruptions, we now investigate the suitability of Dome C Br$_{enr}$ as a proxy for past sea-ice variability.
Previous studies support the use of Br$_{enr}$ in reconstructing past Antarctic sea-ice extent (Vallelonga et
al., 2017; Spolaor et al., 2013b). However, these ice core records were retrieved at coastal sites close to
local source areas, where Br$_{enr}$ values were enriched with respect to sea water mass ratio. To the
contrary, due to its position, Dome C receives atmospheric signals from a vast area of the East Antarctic
sector, which extends from the Indian Ocean to the Ross Sea, potentially giving a reconstruction of
past-sea-ice extent over a broader region.
Our 200-year ice core record shows that Br$_{enr}$ has an average value of 0.5 ± 0.3, meaning that it is
typically depleted at Dome C. This reflects the differences in Na and Br depositions as a function of the
distance from the coast, with Br$_{enr}$ values lower than one recorded in sites which are more than 800 km
far from the coast (Vallelonga et al., 2021).  Further, due to the low snow accumulation at this location
and to the low concentrations of bromine, Br$_{enr}$ values can be influenced by surface snow removal by
wind scurrying, changes in meteorological patterns and changes in wind field. For these reasons, and
with the current state of knowledge, the presented bromine record should be interpreted with caution.
To understand the driving patterns of the Dome C record and its suitability to reconstruct past sea-
ice variability, we compare the Dome C $Br_{enr}$ record with the Southern Annular Mode (SAM) Marshall
index (Marshall, 2003), satellite observations of FYSI extent from the source areas over the period
1979-2012, and with the Law Dome methanesulphonic acid ($MSA_{LD}$) profile (Table 2). SAM describes
the poleward/equatorward movement of the westerly winds that circle Antarctica. When these winds,
known as Southern Westerly Winds (SWW), contract towards Antarctica, the SAM is in its positive
phase, vice-versa it is in its negative phase. The strength of wind patterns likely influences the amount
of sea salt aerosols deposited at Dome C (Crosta et al., 2021), as indicated by the positive correlation
of Br and Na with SAM index (0.41 and 0.61, $p$-value < 0.01, respectively). This is in agreement with
recent findings that highlight a prominent northward flow during the SAM negative polarity at Dome
C (Kino et al., 2021). We did not find any correlation between the SAM index and $Br_{enr}$ values.
We find that for the past decades, $Br_{enr}$ at Dome C is mainly influenced by Br deposition, given the
positive and significant correlation of $Br_{enr}$ with total Br (r = 0.76, $p$-value ≤ 0.01) and the negative
correlation with Na (r =-0.37, $p$-value ≤ 0.01) over the entire record (Table 2). Since gas-phase bromine
is emitted in enhanced concentrations (with respect to sea-water ratio) from sea-salt aerosol derived
from surface blowing snow deposited over FYSI, the $Br_{enr}$ signal at Dome C is likely to be mainly
controlled by emissions and recycling from seasonal sea-ice at the Antarctic coast rather than long-
range air mass transport of sea salt aerosols (Spolaor et al., 2013b).
To test this hypothesis, we compared our record with FYSI extent data during the satellite era (1979-
2012) over the main source areas. As previously stated, most of the back-trajectories that reaches Dome
C from the period 1979 – 2018 came from the WP sector (see Section 3.1). However, we found
significant, but weak, correlations between $Br_{enr}$ and FYSI only with the IO sector (11% of the back-
trajectory points satisfying bromine loading condition) and the RS (21%) with r = 0.35 ($p$-value < 0.1)
and r = 0.3 ($p$-value < 0.1), respectively. In contrast, the closer WP sector does not show any significant
correlation (Table 2). Given the main source areas from back trajectory analysis are located in the WP
(45%, Figure S1), we further investigate this sector by considering the MSA record retrieved from the
Law Dome ice core (hereafter $MSA_{LD}$), located at a coastal site facing the WP (Curran et al., 2003).





$MSA_{LD}$ shows a positive and significant correlation with the past sea-ice extent in the WP sector (r =
0.89, p-value $\leq$ 0.01), but it does not correlate with Dome C $Br_{enr}$, strengthening the idea that Dome C
is influenced by a broader source area than Law Dome (Table 2). Based on these findings, a possible
interpretation is that the IO and RS seasonal sea-ice might have a stronger influence on the Dome C
$Br_{enr}$ profile than WP, due to their 211% and 157% average larger FYSI extent than the one recorded in
the WP, leading to a larger emission of reactive bromine into the atmosphere. Overall, we found a weak,
but significant, correlation between the $Br_{enr}$ record and sea-ice extent in East Antarctica (WP+RS+IO)
(r = 0.35, p-value $\leq$ 0.1).
Nevertheless, we need to consider that overall sea-ice extent in East Antarctica has not undergone
significant changes over the last three decades, with an *inter*-annual variability of ~20%. Moreover,
taking into account the observed $Br_{enr}$ depletion at Dome C and the difficulties in capturing relatively
small sea-ice variabilities due to snow remobilization, changes in meteorological patterns and in wind
fields (Vallelonga et al., 2021), we hypothesize that sea-ice extent variability observed over the last
decades has not been large enough to cause a significant variability in the $Br_{enr}$ signal at Dome C.
However, it cannot be ruled out that when longer periods which extend further back in the past are
considered (e.g. glacial/interglacial transitions), $Br_{enr}$ variations could be used as a qualitative tracer (i.e.
to identify transitions between *large* and *small* FYSI extent) for FYSI variability in East Antarctica.
**4. Conclusions**
In this manuscript we presented the first long-term ice core record of bromine and bromine
enrichment from Dome C (Antarctica). Based on observations and modelling results, we propose that
bromine is effectively preserved within the Antarctic plateau snowpack regardless of the intensity of
the incoming UV-radiation. Furthermore, we find that the change in surface UV-radiation due to ozone
hole formation does not affect the contribution of $H_2O_2$, $NO_3^-$ and $NO_2^-$ to the production of OH radicals
and consequently the dominant OH-driven bromide oxidation channel remains slow. We suggest that
neither of these photochemical mechanisms are likely to take place at Dome C, mostly due to the low
concentration of $H_2O_2$ and $NO_2^-$ as well as the low $^{\bullet}OH$ quantum yield from the $NO_3^-$ photolysis. Lastly,





we did not find any evidence of bromine depletion nor enhancement promoted by volcanic eruptions.
Due to the variety of chemical reactions that can influence bromine preservation within the snowpack,
we suggest the inclusion of site-specific studies to assess to what extent bromine is preserved at different
specific locations, i.e. through the analysis of ·OH precursors ($H_2O_2$, $NO_3^-$ and $NO_2^-$).
Finally, we evaluated whether $Br_{enr}$ at Dome C can be used as a sea-ice proxy. Despite finding
weak – but significant - correlations with the Indian Ocean and Ross Sea sectors (which are the ones
presenting the largest FYSI extents) it is difficult to validate $Br_{enr}$ as an effective proxy for past sea-ice
reconstructions in East Antarctica; this is primarily due to low sea-ice variability observed during the
last 30 years. Future investigations at Dome C need to focus on glacial/interglacial transitions to assess
whether $Br_{enr}$ at Dome C can be used as a qualitative sea-ice tracer over longer timescales.
**Author contribution**
**F.B.:** conceptualization, data curation, formal analysis, investigation, methodology, visualisation,
writing - original draft preparation – **R.P.F.:** data curation, formal analysis, investigation, methodology,
software, visualization, writing – original draft preparation, writing – review & editing – **D.S.:** data
curation, formal analysis, methodology, software, visualisation, writing – original draft preparation,
writing – review & editing – **C.T.:** data curation, methodology, writing – review & editing – **C.S.B-B**:
writing – review & editing – **R.H.R.:**, writing – review & editing – **C.S.:** software, writing – review &
editing – **V.C.:** software, writing – review & editing – **C.B.:** funding acquisition, supervision, writing
– review & editing – **A.S.L.:** investigation, writing – review & editing – **A.S.:** conceptualization,
investigation, resources, supervision, writing – review & editing.
**Acknowledgments**
This publication was generated in the frame of Beyond EPICA. The project has received funding from
the European Union's Horizon 2020 research and innovation programme under grant agreement No.
815384 (Oldest Ice Core). It is supported by national partners and funding agencies in Belgium,
Denmark, France, Germany, Italy, Norway, Sweden, Switzerland, The Netherlands and the United
Kingdom. Logistic support is mainly provided by PNRA and IPEV through the Concordia Station



system.  This publication also benefitted from support by the "Programma Nazionale per la Ricerca in
Antartide" (PNRA, project number PNRA16_00295), by the bilateral international exchange award
Royal  Society  (UK)-CNR  titled:  *Antarctic  sea-ice  history:  developing  robust  ice  core  proxies*
(IEC/R2/202110) awarded to RHR and AS. Logistic support is mainly provided by PNRA and IPEV
through the Concordia Station system. This is Beyond EPICA publication number XX.



**Figures and Tables**
**Figure 1** – Sodium (blue line), bromine (red line) and bromine enrichment (yellow line) ice core record
from 1800 to 2012. Thick lines refer to a smoothed 3-year moving average.

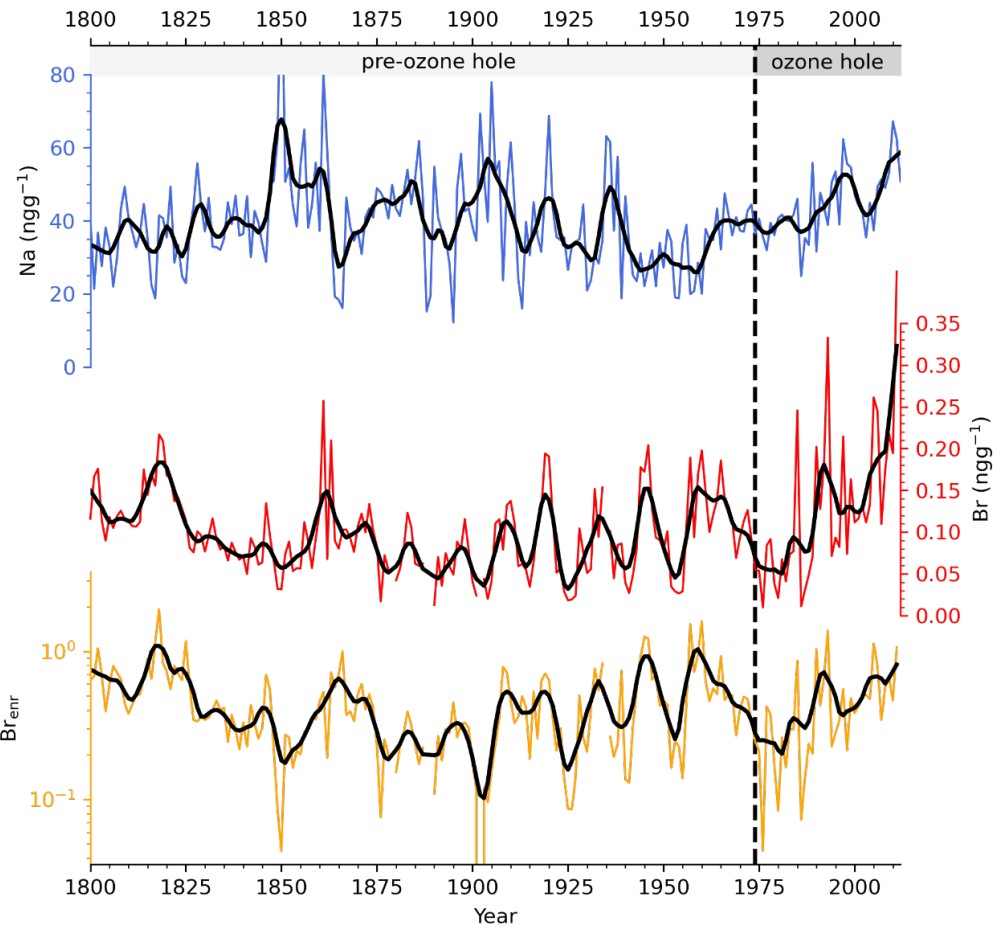









**Figure 2** – Panel a), the wavelength dependent actinic flux (AF) at Dome C for different years (coloured
dots), superimposed with the absorption spectrum of nitrate (violet line), hydrogen peroxide (orange
line) and nitrite (green line). The photolysis rate for the main ˙OH precursors as a function of wavelength
in different years (pre and post the modern ozone hole) is shown for panel b) nitrate, panel c) hydrogen
peroxide and panel d) nitrite.

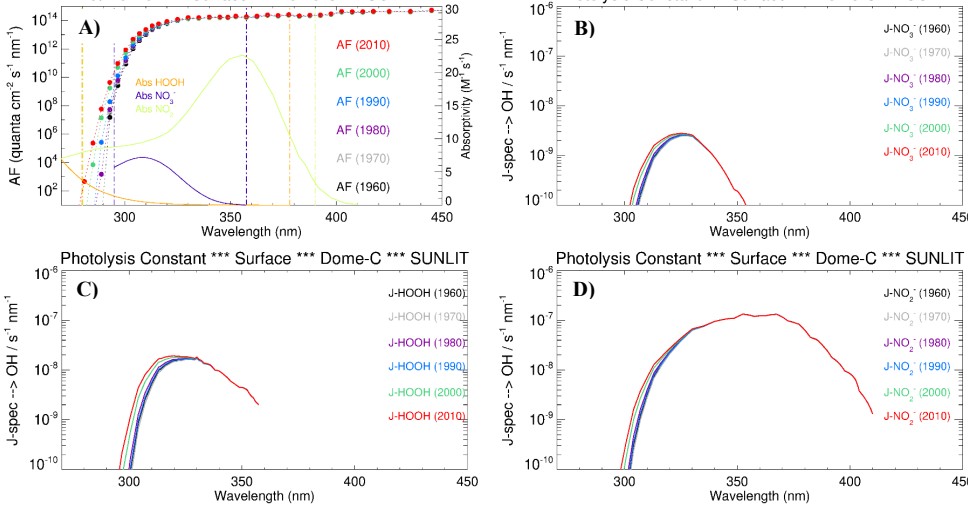

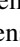
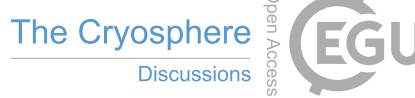

**Figure 3** – Panel a)**:** the photolysis constant of hydrogen peroxide (orange line), nitrate (violet) and
nitrite (green) over the period 1950-2009. Panel b): the $\cdot$OH production rate after the hydrogen peroxide
photolysis (orange), nitrate (violet) and nitrite (green) over the period 1950-2009. Panel c): normalized
photolysis rate for hydrogen peroxide photolysis (orange line), nitrate (violet line) and nitrite (green
line) over the period 1950-2009. The dashed-grey vertical line (1975) represents the beginning of the
ozone-hole period. The horizontal-coloured lines represent the average magnitude during the pre-ozone
hole period (1950-1975).

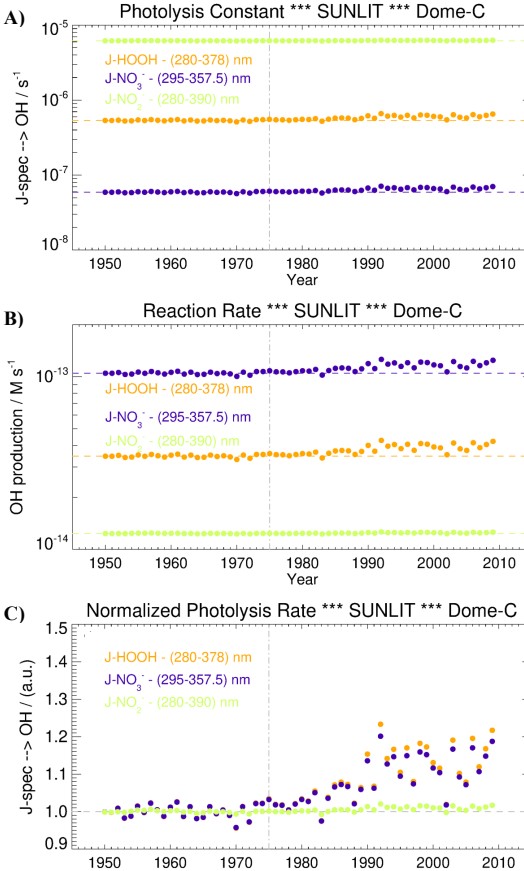




**Table 1** – Summary of the $^{\cdot}$OH quantum yields for $H_2O_2$, $NO_3^-$ and $NO_2^-$, their molar absorptivities and
their concentration at Dome C. *: estimated (more details in the text). [†]: the value is reported from 3.5
m depth.

| Species | $^{\cdot}$OH quantum yield (Chu and Anastasio, 2007, 2005, 2003) | Concentration at Dome C |
|---------|------------------------------------------------------------------|--------------------------|
| $H_2O_2$ | 0.7 | 2 ng g$^{-1}$ (Frey et al., 2006)[†] |
| $NO_3^-$ | 0.0034 at pH = 5 | 110 ng g$^{-1}$ (Spolaor et al., 2021) |
| $NO_2^-$ | 0.020 (T = 260 K, λ = 280 nm) | 0.092* ng g$^{-1}$ |








**Table 2** – Pearson correlations of 3-years moving average of Dome C Br, Na and Br$_{enr}$, Law Dome MSA (MSA$_{LD}$) and seasonal sea-ice extents of the Indian
Ocean (IO), Western Pacific (WP) and Ross Sea (RS) sectors and of the Eastern Antarctic Ocean (EAO=IO+WP+RS), calculated as $FYSI = \overline{Extent_{Sep-Dec}} - $
$Extent_{Feb}$. The moving average is calculated in order to account for a dating error of ~ 3 years. The correlations among the chemical species (Na, Br, Br$_{enr}$ and
NO$_3^-$) were made over the entire record (1800-2012), the correlations with MSA$_{LD}$ were done between 1843-1995, the correlations with SAM (Marshall index)
were done between 1957-2012, the correlations with sea-ice extents were done between 1979-2012. \*\*\*: p-value ≤ 0.01; \*\*: p-value ≤ 0.05; \*: p-value ≤ 0.1.

| Period | Na | Br | Br$_{enr}$ | NO$_3^-$ | MSA$_{LD}$ | SAM | IO | WP | RS | EAO |
|---|---|---|---|---|---|---|---|---|---|---|
| | | 1800-2012 | | | 1843-1995 | 1871-2012 | | 1979-2012 | | |
| Na | 1.0*** | 0.06 | -0.37*** | 0.53*** | -0.04 | 0.61*** | 0.16 | -0.2 | 0.74*** | 0.64*** |
| Br | 0.06 | 1.0*** | 0.76*** | 0.58*** | -0.05 | 0.41*** | 0.36** | -0.24 | 0.44** | 0.48*** |
| Br$_{enr}$ | -0.37*** | 0.76*** | 1.0*** | 0.45*** | 0.05 | 0.18 | 0.35* | -0.22 | 0.3* | 0.35* |
| NO$_3^-$ | 0.53*** | 0.58*** | 0.45*** | 1.0*** | 0.04 | 0.24 | 0.3 | -0.23 | 0.43** | 0.44** |
| MSA$_{LD}$ | -0.04 | -0.05 | 0.05 | 0.04 | 1.0*** | -0.08 | -0.13 | 0.89*** | 0.32 | 0.58** |
| SAM | 0.29*** | 0.47*** | 0.28*** | 0.32** | -0.47*** | 1.0*** | 0.2 | -0.41** | 0.57*** | 0.44** |
| IO | 0.16 | 0.36** | 0.35* | 0.3 | -0.13 | 0.28 | 1.0*** | 0 | -0.06 | 0.42** |
| WP | -0.2 | -0.24 | -0.22 | -0.23 | 0.89*** | -0.14 | 0 | 1.0*** | -0.03 | 0.31* |
| RS | 0.74*** | 0.44** | 0.3* | 0.43** | 0.32 | 0.58*** | -0.06 | -0.03 | 1.0*** | 0.82*** |
| EAO | 0.64*** | 0.48*** | 0.35* | 0.44** | 0.58** | 0.47*** | 0.42** | 0.31* | 0.82*** | 1.0*** |




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
