# Peer review of "200-years ice core bromine reconstruction at Dome C (Antarctica): observational and modelling results"

_The Cryosphere, 2022_

## Author Response (AR1)

**AUTHOR'S RESPONSE TO REFEREE 1**

**Overall the manuscript contributes new and important understanding of bromine preservation and potential environmental influences at Dome C. The manuscript objectives, methodology and results are mostly clear. Minor revision suggestions below:**

We would like to thank the reviewer for the general feedback to our manuscript and for all the suggested implementations/comments.

**Method suggestions:**

**- move equations from the results section to method section. e.g. equation 1. Discuss only results / discussion in results / discussion section.**

We moved eq.1 to the methodological section as suggested (L132-134)

**- move the matlab function description to the method section**

We moved the findchangepts() Matlab function description to the method section (L168-169)

**- move the correlation analysis methods + note the 3-yr moving average (& why this time period was chosen) used for correlation to the methods section - currently in the results / discussion and table 2.**

We moved this part in the method section and we briefly explained why we used the 3-years moving average (L165-168):

*The correlations computed among the different variables of this study (Na, Br, $Br_{enr}$, sea-ice extent data, SAM index) were performed using a 3-year moving average. This choice takes into account both the dating error of the core ($\approx$ 3 years) and the effects of wind erosion on the age distribution of surface snow that spans over more than a year (Picard et al., 2019). To identify any abrupt change point in the records, the findchangepts() Matlab (Mathworks) function was used.*

**Results / Discussion suggestions:**

**- First sentence (line 204) is repeated from methods / add to methods.**

We removed the sentence.

**- Suggest moving Figure S1 to main figures - its your first result that you discuss and you refer to it alot throughout your manuscript**

Thanks for the suggestion. Done

**- Line 214 / Figure 1: the recent increase / peak that you mention look similar to previous increases in Na over the record presented. Recent increase in line with past conditions? The current text is misleading suggesting that the recent abrupt Na increase are unprecedented.**

We agree and we reformulate the sentence (L214-216)

**Line 228 - reference needed after 400km inland**

Reference added

**Line 260: replace regions with wavelengths?**

We replaced regions with wavelength bands.

**Line 263: reference need at end of line**

Reference added

**Line 280: add wavelengths after 340nm**

Done

**Line 398: add a reference to Figure S1 (but suggest moving this to main text)**

Done

**Line 401: sentence unclear. Is the 800km referring to Dome C or a different site. Suggest adding the values from the 800km from coast site for comparison.**

We reformulate the complete sentence (L403-404), and now it should be clearer. In Vallelonga et al., 2021 (see quote below) is reported that Brenr < 1 are usually observed in sites that are located more than 800 km inland from the coast. Considering that Dome C is located 1000 km inland from the coast, our findings are consistent with what has been reported in literature. For the sake of clarity we also changed the sentence at L239-241.

*The different Na and Br spatial snowpack concentration patterns are also reflected in the resulting Brenr and nssBr patterns, which are again very similar for the Arctic and Antarctic. Brenr starts with low values near the coast and gradually increases to the highest values approximately 300-600 km inland. Brenr values then quickly decrease further inland and are less than 1 for all sites greater than 800 km from the coast.*

**Line 407: is the correlation seasonal / annual / inter-annual time period. 3 yr moving average mentioned in table but no reference in methods or discussion. Also recommend stating the resolution / dating error of Dome C and the temporal resolution that can be explored for correlations.**

The correlation was performed using the annual averaged Southern Annular Mode (SAM). More details regarding the 3 yr moving average (L165-168), the age uncertainties (L96-97) and the temporal resolution (L109) are now reported in the methodological section. A full discussion about the shallow core chronology is provided in Spolaor et al., 2021 (e.g. Table S1)

**Line 410: reference for Law Dome MSA record**

Added at L412-413

**Line 418: can you expand on why Br and Br enrich have a different correlation result with SAM?**

We did not deepen our understanding on why Br and Br enr have a different correlation with SAM since our focus was mainly on investigating the suitability of Brenr as a sea-ice proxy for the East Antarctic Sector. However, we hypothesize that the lack of correlation between SAM and Brenr is due to the enhanced transport of Na during the positive SAM phase (SAM and Na are indeed positively correlated), which results into lower Brenr values (since Brenr is defined as Br-to-Na ratio). We believe that it is not worth include it in the main text since it is just a consequence of the positive correlation between SAM and Na.

**Line 426: suggest adding 'defined through back trajectory analysis (Figure S1)'.**

Done

**Minor revision suggestions:**

**- multiple long sentences throughout the manuscript. Suggest splitting these into 2 sentences to make the message clearer. Examples start on: line 10, line 42, line 225, line 228, line 242, line 316**

We edited the suggested and few other sentences accordingly to the reviewer suggestion.

**- abbreviations - need to be consistent with using them. Recommend only adding an acronym if you refer to it more than 5 times.**

We modified some of the acronyms accordingly.

**AUTHOR'S RESPONSE TO REFEREE 2**

**This manuscript concerns the investigation of a shallow firn core drilled in East Antarctica to address the behavior of Br at a low snow accumulation site, evaluating its resistance to post-depositional processes. The manuscript is well written and clear and provides important novelties in the field of halogen glaciochemistry and in the direction of the development of new proxies to reconstruct first year sea ice extension in Antarctica. I am not an expert in photochemistry so I can't really comment on the part dedicated to the different contributions of the sources for the hydroxyl radical in inner East Antarctica, but from what I see the authors made a nice job trying to take into consideration all the relevant aspects and hypotheses. My only real concern is about the importance given to the discussion that tries to link Br records and volcanic activities.**

We are thankful to the reviewer for their feedback on our manuscript and on their comments/suggestions.

**I think that the attempt to discuss your results in relation with volcanic eruptions is not well constrained. I suggest you to remove it or to drastically shorten it. As the authors say it is true that the ice core they investigated ideally contains the traces of a few known volcanic eruptions which had a global impact in the last 200 years, namely Pinatubo, Tambora, Krakatua and others. As the author do not find any anomaly in their Br and Brenr records in this time interval, they argue that volcanic aerosol, in particular acidic species, do not affect the preservation of Br in snow at Dome C. This line of reasoning is definitely interesting but in my opinion the authors are going a little bit too far with speculation here. At first they did not measure any proxy for volcanic eruptions, secondarily if we consider past research on recent volcanic eruptions recorded at Dome C (see for example Castellano et al., 2005, JGR 110:D06114), we see that eruptions occurred in the last 200 years produced glaciochemical signals that are not among the most intense recorded at Dome C. Personally I found the paragraph dedicated to this discussion a little bit overstating. You could consider to drastically shorten this part. About the fact that in West Antarctica there are some evidence of volcanic-related interferences in Br preservation: this is probably related to the fact the event considered in the cited paper was a local Antarctic one, producing a local glaciochemical impact on snow properties that was much stronger that the one produced by eruptions that despite being of global relevance had only limited impact on remote areas such as Dome C. This is definitely an interesting topic to investigate but I feel that in its current state this is not the right study for this. What you can tell here is that from your evidence you can say that in the considered record volcanic eruptions are not impacting Br-related records, but being the considered time interval short and the site notably peculiar, further studies are needed to better investigate this.**

**This is the only relevant issue that I have noted in this manuscript, once the authors have adjusted this, I will recommend its publication.**

We thank the reviewer for this punctual comment and we agree on their suggestion. We edited the text accordingly to highlight that our findings need to be corroborated by other studies. We edited the 3.3 section accordingly (L382-392):

*Nevertheless, we highlight that, due to low snow accumulation and strong wind erosion, not all the volcanic eruptions listed above might be present in our record. Indeed, a previous investigation that compared the sulphate signal from five ice cores drilled 1 meter apart from each other at Dome C showed a bulk probability of 30% of missing volcanic events when a single core is used as the site reference (Gautier et al., 2016). Among the volcanic events embraced by our record, only Krakatua, Cosiguina*

*and UE 1809 were observed in all the previously mentioned five replicate cores, giving us confidence that for these eruptions the volcanic fingerprint is present also in our record. However, we acknowledge that a proxy-based volcanic reconstruction is missing for our core and, considering the strong spatial variability observed at Dome C, further and more specific studies are needed to investigate the impact of large inter-hemispherical volcanic eruptions on the preservation of bromine in the snowpack.*

**Below a few specific comments**

Below we provide specific responses to each of the reviewer individual comments

**Line60:maybe better "inversely" than "linearly"?**

Done

**Line103: why snow accumulation of the 2006-2013 interval is missing the standard deviation?**

We re-checked the data and we changed some of the snow accumulation values with more updated ones (we added a new reference).

Frezzotti M, Scarchilli C, Becagli S, Proposito M, Urbini S. A synthesis of the Antarctic surface mass balance during the last 800 yr. The Cryosphere. 2013 Feb 20;7(1):303-19.

Unfortunately, the standard deviation is not provided in neither of the cited manuscript. However, the evolution of snow accumulation (in cm) is provided in Genthon et al., 2016. The conversion in kg m$^{-2}$ yr$^{-1}$ is done using an average surface snow density of 320 kg m$^{-3}$.

Genthon, C., Six, D., Scarchilli, C., Ciardini, V., and Frezzotti, M.: Meteorological and snow accumulation gradients across Dome C, East Antarctic plateau, International Journal of Climatology, 36, 455-466, 2016.

**Line116: it is not common to perform ICP-MS analyses on samples that were not acidified, as this is uncommon, could you spend some words to better explain and justify this choice? I am also asking if the same treatment was followed for standards**

Usually the acidification of molten ice/snow samples is performed when we want to dissolve particle-bound elements, such as iron (e.g. Burgay et al., 2021). In the case of this study, we did not acidify the samples because we were only interested in the water-soluble Br and Na fractions. Consistently, also the standards were not acidified and the overall procedure (cleaning of vials, etc…) did not use any acid. Lastly, the acidification of the samples might led to analyte loss when halogens are targeted (Flores et al., 2022). We add additional text at L114-115 and L125-126 for clarification.

Burgay, F., Spolaor, A., Gabrieli, J., Cozzi, G., Turetta, C., Vallelonga, P., & Barbante, C. (2021). Atmospheric iron supply and marine productivity in the glacial North Pacific Ocean. *Climate of the Past*, *17*(1), 491-505.

Flores, E. M., Mello, P. A., Krzyzaniak, S. R., Cauduro, V. H., & Picoloto, R. S. (2020). Challenges and trends for halogen determination by inductively coupled plasma mass spectrometry: a review. *Rapid Communications in Mass Spectrometry*, *34*, e8727.

**Line225-228: as this is not something coming from this manuscript, I would expect to find some references here**

The statements presented from L225 to L228 refers to Vallelonga et al., 2021, which is now explicitly cited in the text.

**Line263-165: could you provide a rough estimate of this increase? Just to better understanding the intensity of the change**

The UV-forcing roughly increase 10 times (see Figure 1 in Spolaor et al., 2021). We add this value (and associated reference) at L265.

Spolaor, A., Burgay, F., Fernandez, R. P., Turetta, C., Cuevas, C. A., Kim, K., ... & Saiz-Lopez, A. (2021). Antarctic ozone hole modifies iodine geochemistry on the Antarctic Plateau. *Nature communications*, *12*(1), 1-9.

**Line365: what about coastal sites where snow acidity is also enhanced by biogenic marine emissions?**

This is a great point. We decided not to discuss in details because it is beyond the scope of the manuscript (i.e. Dome C is not a coastal site), but we added this suggestion at L361-362.

**Line367: please see my comment about this part above**

This section has been implemented accordingly with the precious suggestions given by the reviewer. See our previous answer for more details.

**Line 442-443: maybe better adding a reference for this here**

Done at L429-430. In general, the sea-ice extent data were also retrieved from the NASA Earth Science portal (https://earth.gsfc.nasa.gov/) and at the National Snow and Ice Data Center portal (http://nsidc.org) as stated at L160-162. To better highlight how sea-ice has changed over the investigated time period, we added a new figure (Figure S3) in the supplementary material.

**Line 469: I would reformulate in a more conservative way, something like: "Future investigations at Dome C need to focus on glacial/interglacial transitions to assess whether Brenr at Dome C is somehow related to large scale variations of sea-ice extensions."**

Thanks for the suggestion. We changed accordingly